# Attribution-Driven Adaptive Token Pruning for Transformers

**Yaoyao Yan**[1]    **Hui Yu**[2]    **Weizhi Xu**[1*]

[1]School of Information Science and Engineering, Shandong Normal University
[2]Business School, Shandong Normal University

yanyy@stu.sdnu.edu.cn
huiyu0117@sdnu.edu.cn
xuweizhi@sdnu.edu.cn [*]

## Abstract

Transformers have been widely adopted in natural language processing, computer vision, and other domains due to their exceptional performance across a variety of tasks. However, the computational cost of Transformers is prohibitively high, particularly when handling long input sequences, significantly increasing both training and inference time. Although various token pruning methods have been proposed to reduce the computational burden of Transformers, most approaches overlook critical differences in sequences in terms of length and complexity, leading to suboptimal compression efficiency.

In this paper, we propose AD-TP, an Attribution-Driven Adaptive Token Pruning method designed to retain only the most informative tokens. We analyze the performance of using accumulated attention values to measure token importance and find that attention values do not accurately reflect the actual contribution of each token to text understanding. Additionally, we observe significant variations in the length and complexity of different sequences within the dataset. Based on these insights, we adopt Integrated Gradients to evaluate token importance and introduce a lightweight adaptive token retainer module that dynamically generates pruning configurations for each input sequence. In addition, we incorporate both teacher supervision and self-supervised learning objectives to enhance the training efficiency, accuracy, and robustness of the model.

Experiments conducted on GLUE, SQuAD, and 20News demonstrate that AD-TP outperforms state-of-the-art token pruning and model compression methods in both accuracy and computational efficiency. On GLUE, AD-TP reduces FLOPs by an average of 7.8× while improving performance by 0.6%.

## 1   Introduction

Transformers such as GPT and BERT are central to Natural Language Processing (NLP) due to their strong language modeling capabilities [1, 2]. However, their self-attention mechanism scales quadratically with sequence length, resulting in high computational cost and latency. For instance, processing 512 tokens requires over 1.5 billion Floating-Point Operations (FLOPs), limiting real-time or edge deployment. These challenges have motivated increasing interest in model compression techniques to enhance efficiency without sacrificing performance.

To address the computational bottlenecks inherent in large-scale Transformer models, researchers have explored several major compression techniques, including pruning [3–5], quantization [6–8],

---

[*]Corresponding author: Weizhi Xu

low-rank decomposition [9], and knowledge distillation [10, 11]. Among these, token pruning has received increasing attention due to its hardware-agnostic design and its ability to adaptively retain informative tokens based on sequence content [12].

Recent efforts have applied token pruning to compress Transformer models and accelerate inference. Early work such as PoWER-BERT [13] adopted layer-wise embedding pruning and achieved a 4.5× speedup, but required retraining under multiple constraints. LAT [14] addressed this by using LengthDrop to generate sub-models via evolutionary search, yet it pruned all sequences to a fixed length, leading to under-pruning or over-pruning depending on the sequence. Subsequent methods like SpAtten [15] and TR-BERT [16] introduced length-aware or content-aware strategies, but either lacked semantic adaptability or incurred high training costs. LTP [17] further improved adaptiveness by learning per-layer thresholds to prune unimportant tokens, but still applied fixed thresholds during inference.

Despite these advances, two key challenges remain: (i) most approaches rely on attention scores to assess token importance, which may not accurately reflect true contribution; and (ii) many pruning strategies adopt static configurations, ignoring variations in sequence complexity. Addressing these limitations requires a more accurate importance estimation method and a mechanism that dynamically adjusts pruning decisions based on each sequence.

To address the above limitations, we propose AD-TP, an Attribution-Driven Adaptive Token Pruning method in this work. We analyze the effectiveness of cumulative attention scores in evaluating token importance, and find that attention-based measures fail to reliably identify the tokens that should be preserved during pruning. To overcome this, we adopt an attribution method based on Integrated Gradients (IG), which more accurately captures which tokens truly drive the predictions of the model. Moreover, we observe substantial variation in the complexity of different sequences. To accommodate this, we design an adaptive token retainer module, which integrates an Adaptive Retention Ratio Predictor (ARP) and a Token Saliency Predictor (TSP) to dynamically determine the token retention ratio for each sequence.

Specifically, this work makes the following major contributions:

- We propose AD-TP, which introduces a lightweight adaptive token retainer that dynamically determines the retention ratio based on sequence complexity, thereby significantly reducing computational cost.

- We introduce a novel attribution-based token importance estimation approach, which leverages IG to quantify the relationship between model predictions and sequence features, enabling a more accurate assessment of the contribution of each token to the output.

- We design a knowledge distillation framework with dual normalization to align attribution features between teacher and student models, and incorporate self-supervision to enhance token-level reasoning in the student model.

- Through comprehensive experimental evaluations, AD-TP achieves average FLOPs reductions of 7.02× and 7.37× on the GLUE benchmark using 6-layer and 12-layer Transformers, respectively, without sacrificing accuracy. Furthermore, AD-TP also demonstrates superior performance on long-sequence tasks.

## 2    Related Work

**Token Pruning**    Existing token pruning methods can be broadly classified into two categories based on their retention strategies. The first type relies on attention-based scoring, using attention weights to estimate token importance. For instance, SpAtten [15] ranks tokens based on attention scores, while PoWER-BERT [13] and LTP [17] perform sequence pruning based on attention representations using fixed or learnable thresholds. STTABT [18] traces attention backward across layers to assess token contribution, and LAD [19] transfers attention knowledge for dynamic pruning in small models. The second type uses prediction-based pruning, introducing predictors before each Transformer layer to estimate token retention. TR-BERT [16] employs reinforcement learning, and Transkimmer [20] uses a two-layer MLP to score tokens. These approaches explicitly model importance and offer better pruning accuracy and flexibility. The working mechanism of Transformer models and the underlying principles of token pruning are detailed in Appendix A.1.

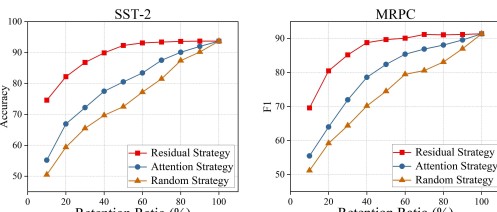
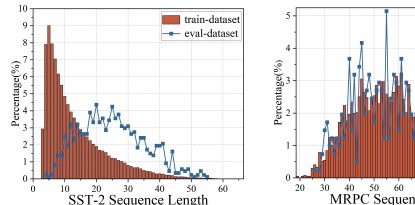

Figure 1: Model performance under different token importance estimation strategies and token retention ratios.

Figure 2: Sequence length distribution statistics for training and validation sets.

**Attribution Method Based on Integrated Gradients**   Attribution methods assign model predictions to sequence features using saliency metrics like gradients [21]. Recent work applies attribution to analyze attention patterns in Transformers [22–24]. Among these, IG offers stable and continuous estimates by integrating gradients along input paths, and has been widely adopted for interpreting deep models in NLP [25]. In aspect-based sentiment analysis, IG effectively identifies tokens relevant to specific aspects [26]. It has also been used with pre-trained models like BERT and RoBERTa to extract features important for classification, thereby improving performance [27]. IG further reveals how linguistic constructs such as negation and conjunctions influence predictions in text classification tasks [28]. To address the discreteness of embedding spaces, Roy et al. [29] introduced the Unified Discretized IG method, enhancing interpretability for large language models such as BERT. The fundamental principles of the IG are detailed in Appendix A.2.

## 3   Motivation

In this section, we analyze the strengths and limitations of attention-based token importance estimation, examine variations in sequence length and complexity, and summarize the challenges in designing adaptive token pruning methods.

### 3.1   Impact of Token Importance Estimation Strategies on Model Performance

Token importance estimation is crucial for effective pruning. Most existing methods rely on attention mechanisms to assess token importance; however, such mechanisms focus on token-to-token interactions and tend to overlook the independent semantic contribution of individual tokens [30]. To evaluate the effectiveness of different estimation strategies, we compare three approaches on the SST-2 and MRPC datasets: a random strategy (as a lower bound), an attention strategy, and a residual strategy (as an upper bound). As shown in Fig. 1, the attention-based approach still leaves substantial room for improvement in pruning performance, indicating a discrepancy between attention scores and the actual semantic contribution of tokens. To address this issue, we propose an attribution-based method that leverages IG to quantify the marginal contribution of each token to the model output, enabling more accurate and interpretable token selection.

### 3.2   Limitations of Fixed and Learnable Thresholds

Most pruning frameworks adopt either fixed or globally learned thresholds, which fail to account for the variability in sequence length (Fig. 2) and complexity (Fig. 3) across and within datasets. Although LTP [17] improves flexibility by learning layer-wise thresholds during training, these thresholds remain static at inference time and lack instance-level adaptability. To overcome this constraint, we propose a dynamic pruning mechanism that adjusts the pruning ratio for each sequence based on estimated token saliency and structural complexity.

### 3.3   Challenges

Despite the significant potential of attribution-driven adaptive token pruning methods in improving model efficiency while preserving predictive performance, several challenges remain in practical implementation. First, the IG method must remain stable and effective under conditions of sparse token

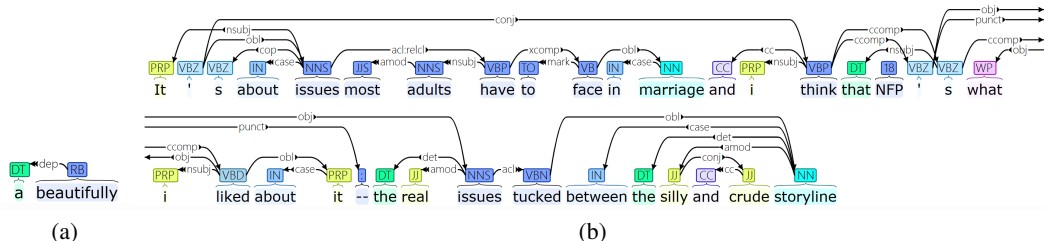

(a)                                    (b)

Figure 3: Syntactic dependency analysis of sample sequences in SST-2.

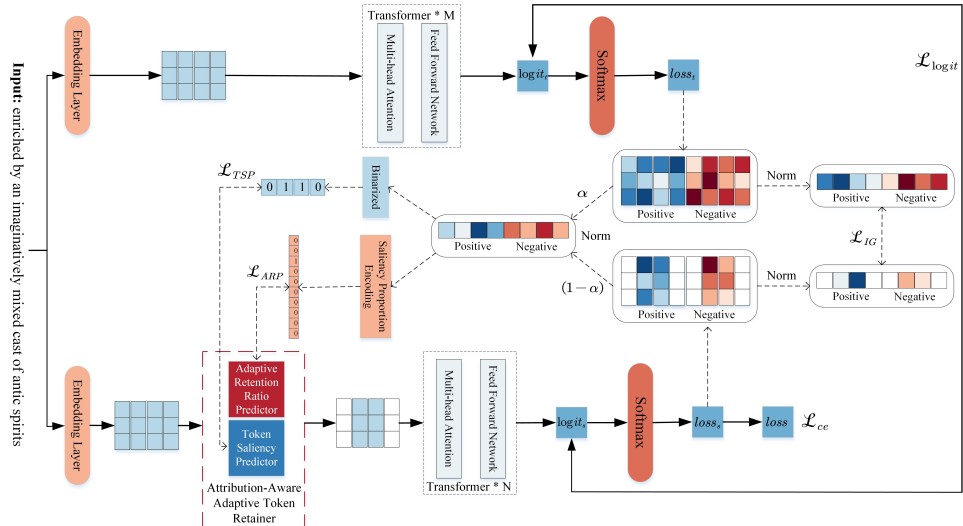

Figure 4: Overview of the AD-TP framework. The red dashed box represents the Attribution-Aware Adaptive Token Retainer, which consists of ARP and TSP.

retention. Second, the adaptive token retainer must be sufficiently lightweight to avoid introducing additional inference overhead. Third, given that the training process involves knowledge distillation, token removal inevitably leads to information loss and results in representation misalignment between the student and teacher models.

## 4 Attribution-Driven Adaptive Token Pruning

In this section, we present the AD-TP method in detail. The overall framework is illustrated in Fig. 4. This approach prunes sequences by introducing an Attribution-Aware Adaptive Token Retainer, which consists of an ARP and a TSP. AD-TP leverages IG to assess token saliency and combines teacher supervision with self-supervised to guide pruning. Additionally, an attribution-based distillation strategy is employed to enhance model representation and performance.

### 4.1 Problem Definition

Given an input token sequence $x = \{x_1, x_2, ..., x_n\}$, where $x_i$ denotes an input token and $n$ is the sequence length. Transformer first maps the token sequence into a $d$-dimensional embedding sequence $E = \{e_1, e_2, ..., e_n\}$ through an embedding layer. Each $e_i$ represents the embedding of token $x_i$.

### 4.2 Attribution-Aware Adaptive Token Retainer

As illustrated in Fig. 5, the Attribution-Aware Adaptive Token Retainer consists of two main components: ARP on the left estimates the complexity of the sequence and determines the retention ratio $\rho_r$,

while TSP on the right evaluates the importance of each token in the embedding sequence $E$. Based on the retention ratio, this module dynamically generates a sparse representation, enabling adaptive pruning of the sequence.

In the ARP module, the embedding of the special classification token [CLS], denoted as $x_g \in \mathbb{R}^d$, is first used as a global representation of the input sequence. This global token is then passed through a linear layer and an argmax function $\mathcal{G}$ to produce a one-hot vector that selects $\rho_r$ from a candidate list $L_\rho$. The process is defined as:

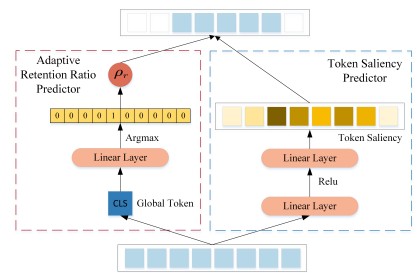

$$x_g = e_{[\text{CLS}]}, \quad x_\rho = linear\,(x_g) \qquad (1)$$
$$one\_hot = \mathcal{G}\,(x_\rho), \quad \rho_r = one\_hot \cdot L\rho \qquad (2)$$

where $e_{[\text{CLS}]} \in \mathbb{R}^d$ denotes the final-layer embedding of the [CLS] token, and $\mathcal{G}$ is the argmax activation function, which maps the logits $x_\rho$ to a one-hot vector indicating the selected retention ratio.

Figure 5: Attribution-Aware Adaptive Token Retainer.

To identify key tokens under the retention ratio $\rho_r$, the TSP assigns a saliency score $S \in \mathbb{R}^n$ to each token. Each embedding from $E$ is passed through a linear layer, ReLU activation, and another linear projection to produce the score:

$$S = linear_2\,(ReLU\,(linear_1\,(E))) \qquad (3)$$

Based on the adaptive retention ratio $\rho_r$ and the token saliency scores $S$, we select the top $\rho_r\%$ most important tokens to construct an adaptive sparse sequence for subsequent computation.

## 4.3 Token Saliency Estimation Based on Integrated Gradients

In AD-TP, IG are used to supervise both token pruning and knowledge distillation. IG computes attribution scores by integrating gradients along a path from a baseline input (e.g., [PAD]) to the actual input, and is formally defined as follows:

$$IG_{ij}\,(F^c, E) = \left(e_{ij} - e'_{ij}\right) \times \sum_{k=1}^{m} \frac{\partial F^c\left(E' + \frac{k}{m} \times (E - E')\right)}{\partial e_{ij}} \times \frac{1}{m} \qquad (4)$$

where $m$ is the number of integration steps, $F^c$ is the predicted score for class $c$, and $e_{ij}$ is the $j$-th feature of the $i$-th token. The baseline $e'_{ij}$ is set as the [PAD] embedding. In our setup, $m = 1$.

To reduce the influence of low-information dimensions, the teacher model retains only the top-$K$ IG dimensions per token and computes their L2 norm. To minimize training complexity and avoid noise, the student model retains all dimensions.

$$a_{t,c}^i = \|\,\text{TopK}\,(\,\text{IGapprox}\,_i\,(F_{t,c}, E_t))\,\|_2, \quad a_{s,c}^i = \|\,(\,\text{IGapprox}\,_i\,(F_{s,c}, E_s))\,\|_2 \qquad (5)$$

where $a_{t,c}^i$ and $a_{s,c}^i$ represent the attribution scores of token $x_i$ with respect to class $c$ in the teacher and student models, respectively. $F_{t,c}$ and $F_{s,c}$ are the corresponding prediction functions, and $E_t$ and $E_s$ are the input embeddings.

## 4.4 Joint Supervision Based on Teacher and Self-Supervision

In this section, we describe how to train the Attribution-Aware Adaptive Token Retainer based on attribution scores. The training process integrates teacher supervision and self-supervised. We perform a weighted fusion of the attribution scores $a_{t,c}^i$ from the teacher model and $a_{s,c}^i$ from the student model to construct a unified supervision signal:

$$a_c^i = (1 - k)a_{t,c}^i + k a_{s,c}^i, \quad k = \min\,(1.0, epoch/num\_epoch) \qquad (6)$$

where, $epoch$ is the current step and $num\_epoch$ is the total number of training epochs. In the early stage of training, due to the limited attribution capability of the student model, the supervision primarily relies on the teacher model. As training progresses, the scores of the student model are gradually incorporated to enhance self-supervised learning.

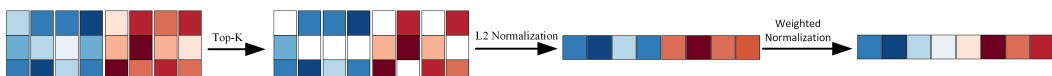

Figure 6: Dual normalization process of attribution scores.

Since the student model retains fewer tokens after pruning, its attribution scores are not directly comparable to those of the teacher model. To address this, we introduce a dual normalization strategy to ensure scale alignment and highlight relative importance. As illustrated in Fig. 6, L2 normalization is first applied to eliminate inter-dimensional disparities, followed by sequence-level normalization to convert scores into relative weights.

$$a_c^l = \|a_c\|_2, \quad a_c^{g,i} = \frac{a_c^{l,i}}{\sum_{i=1}^n a_c^{l,i}} \tag{7}$$

To train the ARP, we use the average saliency score of each sample as a threshold. Tokens exceeding this value determine the target retention ratio, which is then mapped to a candidate list index $y$ as the supervision signal.

$$a_{avg} = \frac{\sum_{i=1}^n a_c^{g,i}}{n}, \quad y = \text{round}\left(l \times \frac{q}{n}\right) \tag{8}$$

where, $q$ is the number of tokens above the mean saliency score, $n$ represents the total number of tokens in the sequence that participate in the saliency evaluation, and $l$ is the size of the candidate ratio list. Adjusting $l$ allows the model to adapt to different computational budgets. The ARP loss is defined as:

$$\mathcal{L}_{ARP} = \mathcal{L}_{CE}\left(x_\rho, y\right) \tag{9}$$

where, $\mathcal{L}_{CE}$ is the cross-entropy loss, and $\mathcal{L}_{ARP}$ encourages $x_\rho$ to match the target index in the candidate list.

Meanwhile, we train the TSP module to estimate token saliency scores. The fused attribution scores $a_c^{g,i}$ are ranked, and the top $\rho\%$ tokens are labeled as 1, while the remaining tokens are labeled as 0 to construct a binary supervision signal $BA$. The loss function is defined as follows:

$$\mathcal{L}_{TSP} = \frac{1}{n} \sum_{i=1}^n \mathcal{L}_{BCE}\left(s_i, BA_i\right) \tag{10}$$

where $s_i$ denotes the predicted saliency of the $i$-th token in the sequence, obtained from Equation (3). $\mathcal{L}_{BCE}$ represents the binary cross-entropy loss function.

## 4.5 Overall Objective

In this section, we describe the overall training objective of AD-TP. To ensure a smoother overall distillation process, we also apply a dual normalization strategy to the attribution scores of both the teacher and student models:

$$a_{t,c}^l = \|a_{t,c}\|_2, \quad a_{s,c}^l = \|a_{s,c}\|_2 \tag{11}$$

$$a_{t,c}^{q,i} = \frac{a_{t,c}^{l,i}}{\sum_{i=1}^n a_{t,c}^{l,i}}, \quad a_{s,c}^{q,i} = \frac{a_{s,c}^{l,i}}{\sum_{i=1}^n a_{s,c}^{l,i}} \tag{12}$$

To represent attribution globally, per-class vectors $a_{t,c}^g$ and $a_{s,c}^g$ are concatenated across $C$ classes:

$$A_t = \Big\|_{c=1}^C a_{t,c}^g, \quad A_s = \Big\|_{c=1}^C a_{s,c}^g \tag{13}$$

where $\|$ denotes the concatenation operation, which aggregates the attribution scores across all categories along a specified dimension. To mitigate pruning-induced loss, an attribution distillation loss minimizes the L2 distance between teacher and student representations:

$$\mathcal{L}_{IG} = \left\|A^t - A^s\right\|_2 \tag{14}$$

The final training objective combines classification loss, logits-based distillation, attribution loss, and token retainer supervision:

$$\mathcal{L} = (1 - \alpha)\mathcal{L}_{CE} + \alpha\mathcal{L}_{logit} + \beta\mathcal{L}_{IG} + \gamma\left(\mathcal{L}_{ARP} + \mathcal{L}_{TSP}\right) \tag{15}$$

Hyperparameters $\alpha$, $\beta$, and $\gamma$ control the contribution of each term.

# 5 Experiments

**Datasets and Evaluation Metrics.** We evaluate AD-TP on 8 tasks from the GLUE benchmark and extend the evaluation to two long-text datasets: SQuAD v2.0 [31] and 20News [32]. Each GLUE task adopts task-specific evaluation metrics; SQuAD v2.0 is evaluated using the F1 score, while 20News uses accuracy. In addition, inference efficiency is assessed using FLOPs, a hardware-agnostic metric that reflects the computational cost of model inference.

**Implementation Details.** All experiments are implemented in PyTorch with Huggingface Transformers on an NVIDIA RTX 3060. The teacher is a 12-layer BERT-base, and the student uses either 6 or 12 layers. We tune learning rates and distillation weights ($\alpha$, $\beta$, $\gamma$) across defined ranges. All hyperparameter configurations and dataset-specific settings are provided in Appendix A.3.

**Existing Methods for Comparison.** To evaluate AD-TP comprehensively, we compare it with representative pruning methods and other model compression techniques. PoWER-BERT [13] uses progressive word-vector elimination; LAT [14] dynamically adjusts sequence length via LengthDrop; and LTP [17] learns attention-based pruning thresholds. Transkimmer [20] employs token-level predictors, while ToP [33] combines ranking distillation with coarse-to-fine pruning. We also include DistilBERT [34] and CoFi [35] as baselines for distillation and structured pruning. Unlike these approaches, AD-TP adaptively predicts retention ratios based on sequence complexity, enabling more flexible and efficient compression. It can also be integrated with orthogonal techniques for further gains.

## 5.1 Main Results

Table 1: Comparison of AD-TP and mainstream pruning methods on GLUE in terms of accuracy and FLOPs. T and S denote the teacher and student models, respectively. All baseline results are from ToP [23], except BERT-base (T) and BERT6 (S). Bold and underlined values indicate the best and second-best results.

| Model | Method | CoLA | | RTE | | QQP | | MRPC | | SST-2 | | MNLI | | QNLI | | STS-B | |
|---|---|---|---|---|---|---|---|---|---|---|---|---|---|---|---|---|---|
| | | Matthews | FLOPs | Acc. | FLOPs | Acc. | FLOPs | F1 | FLOPs | Acc. | FLOPs | Acc. | FLOPs | Acc. | FLOPs | Pearson | FLOPs |
| BERT-base (T) | - | 60.3±0.7 | 1.00× | 69.7±0.5 | 1.00× | 91.5±0.9 | 1.00× | 91.4±0.3 | 1.00× | 93.7±0.5 | 1.00× | 84.9±0.4 | 1.00× | 91.7±0.5 | 1.00× | 89.0±0.2 | 1.00× |
| BERT6 (S) | - | 51.2±1.1 | 1.00× | 66.1±0.4 | 1.00× | 90.4±0.3 | 1.00× | 89.2±0.8 | 1.00× | 91.0±0.8 | 1.00× | 81.7±0.9 | 1.00× | 89.3±0.5 | 1.00× | 87.8±0.5 | 1.00× |
| PoWER-BERT | Atten-value | 52.3 | 4.50× | 67.4 | 3.40× | 90.2 | 4.50× | 88.1 | 2.70× | 92.1 | 2.40× | 83.8 | 2.60× | 90.1 | 2.00× | 85.1 | 2.00× |
| LAT | Atten-value | - | - | - | - | - | - | - | - | 92.8 | 2.90× | 84.4 | 2.80× | - | - | - | - |
| LTP | Atten-value | 52.3 | 8.66× | 63.2 | 6.84× | 90.4 | 7.44× | 87.1 | 6.02× | 92.3 | 3.59× | 83.9 | 3.74× | 89.3 | 3.91× | 87.5 | 5.25× |
| ToP | Atten-value | **60.5** | 9.62× | 70.0 | **7.10×** | 91.2 | **8.04×** | 89.2 | 6.32× | 93.5 | 3.82× | 84.7 | 4.27× | **90.6** | 4.35× | 87.6 | 5.32× |
| AD-TP6 | IG | 55.3±0.4 | 13.78×±0.2 | 70.1±0.7 | 4.25×±0.4 | 90.4±0.5 | 7.05×±0.3 | 89.8±0.9 | 6.83×±0.3 | 92.0±0.7 | 7.98×±0.4 | 84.9±1.0 | 5.45×±0.7 | 89.5±0.5 | 4.17×±0.3 | 88.2±0.6 | 6.63×±0.7 |
| AD-TP12 | IG | 56.9±0.6 | **13.89×±0.4** | 71.2±0.7 | 4.89×±0.7 | 91.4±0.6 | 7.85×±0.5 | 90.8±0.6 | 6.97×±0.4 | 94.7±0.3 | 8.45×±0.6 | 85.4±0.4 | 5.69×±0.2 | 90.2±0.6 | 4.52×±0.4 | 88.7±0.4 | 6.66×±0.3 |

Table 2: Comparison of Token Pruning Methods on Long-Sequence Tasks.

| Model | Method | SQuADv2.0 | | 20News | |
|---|---|---|---|---|---|
| | | F1 | FLOPs | Acc. | FLOPs |
| BERT-base | - | 77.1±0.3 | 1.00x | 86.7±0.7 | 1.00x |
| BERT6 | - | 73.8±0.8 | 1.00x | 85.0±1.2 | 1.00x |
| Transkimmer | Prediction | 75.7 | 4.67x | 86.1 | 8.11x |
| PoWER-BERT | Attention | - | - | 86.5 | 2.91x |
| LTP | Attention | 75.6 | 3.10x | 85.2 | 4.66x |
| ToP | Attention | 75.9 | 4.12x | 87.0 | 8.26x |
| AD-TP6 | IG | **76.2±0.6** | 5.07x±0.2 | **87.3±0.5** | 8.74x±0.5 |

Table 3: Comparison with Distillation and Structured Pruning.

| Model | CoLA | | MRPC | | 20News | |
|---|---|---|---|---|---|---|
| | Matt. | FLOPs | F1 | FLOPs | Acc. | FLOPs |
| BERT6 | 51.2±1.1 | 1.00x | 89.2±0.8 | 1.00x | 85.0±1.2 | 1.00x |
| DistilBERT6 | 49.0 | 2.00x | 86.9 | 2.00x | 85.8 | 2.00x |
| CoFi6 | 38.0 | 9.10x | 86.3 | **7.70x** | 85.9 | 5.90x |
| AD-TP6 | **55.3±0.4** | 13.78x±0.2 | **89.8±0.9** | 6.83x±0.3 | **87.3±0.5** | 8.74x±0.5 |
| BERT-base12 | 60.3±0.7 | 1.00x | 91.4±0.3 | 1.00x | 86.7±0.7 | 1.00x |
| CoFi12 | 39.8 | 9.10x | 90.0 | 4.00x | 86.4 | 7.70x |
| AD-TP12 | 56.9±0.6 | 13.89x±0.4 | 90.8±0.6 | 6.97x±0.4 | 88.6±0.3 | 8.82x±0.2 |

**Comparison with Token Pruning Methods.** We evaluate AD-TP on the GLUE benchmark and two long-sequence datasets. As shown in Table 1, AD-TP6 consistently improves both accuracy and efficiency over the unpruned student model. AD-TP12 achieves a 7.37× FLOPs reduction compared to the teacher model while maintaining competitive performance, and outperforms the state-of-the-art ToP method on several tasks. This highlights the advantage of IG over attention-based metrics in accurately estimating token saliency. On long-sequence tasks (Table 2), AD-TP6 delivers significant gains. Notably, it surpasses BERT-base in accuracy on 20News while reducing FLOPs by 8.74×.

**Comparison with Other Model Compression Methods.** We compare AD-TP with DistilBERT and CoFi using both 6-layer and 12-layer BERT models (Table 3). AD-TP achieves significantly higher FLOPs reduction than DistilBERT (e.g., 8.7× vs. 2.0× for BERT6), while maintaining or

improving accuracy. Compared to CoFi, AD-TP consistently delivers better performance under similar or higher compression ratios. Notably, CoFi6 suffers a marked accuracy drop, whereas AD-TP6 preserves model performance. These results demonstrate that AD-TP achieves efficient and robust compression across both compact and large-scale Transformer models.

## 5.2 Ablation study

**Impact of the Teacher Model and Loss Components.**
We conduct ablation studies on 5 variants of AD-TP to evaluate the contribution of each distillation component: removing (1) the teacher model, (2) the cross-entropy loss $\mathcal{L}_{CE}$, (3) the logits-based loss $\mathcal{L}_{logit}$, (4) the attribution distillation loss $\mathcal{L}_{IG}$, and (5) the retainer loss ($\mathcal{L}_{ARP} + \mathcal{L}_{TSP}$). As shown in Fig. 7, removing any component degrades performance, with the absence of the teacher model or retainer loss causing the most significant drop (7–8%). These results confirm the essential role of both the teacher model and the Attribution-Aware Adaptive Token Retainer in guiding effective pruning and preserving model accuracy.

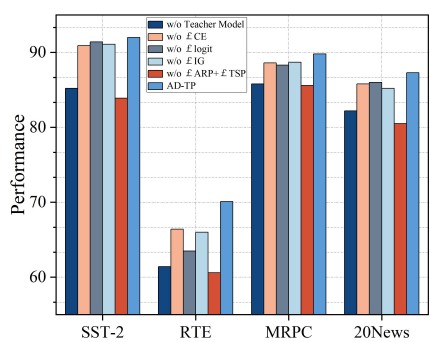

Figure 7: Ablation study of distillation components.

**Impact of the ARP.** To assess the effectiveness of our adaptive pruning strategy, we compare three methods: (i) a static retention ratio fixed at 0.3; (ii) a random ratio selected from a predefined candidate list with the same average; and (iii) our proposed adaptive method (AD-TP). As shown in Table 4, all methods share the same architecture and differ only in how the retention ratio is determined. Results show that the adaptive strategy consistently outperforms the others under comparable FLOPs, demonstrating the benefit of adjusting pruning strength based on sequence complexity. Additional experimental results on more datasets are provided in Appendix A.4.

## 5.3 Effectiveness Under Different FLOPs Budgets

We evaluate the effectiveness of the AD-TP method under varying FLOPs budgets and compare it with two token pruning approaches: LTP and ToP. To ensure a fair comparison, we utilize the official implementations of LTP and ToP and apply grid search to optimize their hyperparameters so that each method achieves its best performance under a given FLOPs constraint. As shown in Fig. 8, AD-TP consistently outperforms both baselines across different tasks under varying FLOPs constraints. On the MRPC, under a 60% relative FLOPs constraint, AD-TP achieves 3% higher F1 score than LTP and 2% higher than ToP. Under the same F1 score (87) constraint, AD-TP reduces FLOPs by approximately 34% compared to LTP and 9% compared to ToP.

Table 4: Ablation study of the adaptive token retainer.

| Model | SST-2 | | 20News | |
|---|---|---|---|---|
| | Acc. | $\rho$ | Acc. | $\rho$ |
| Static | 91.2 | 0.30 | 85.7 | 0.30 |
| Random | 90.5 | 0.30 | 82.5 | 0.30 |
| Adaptive | **92.0** | 0.29 | **87.3** | 0.27 |

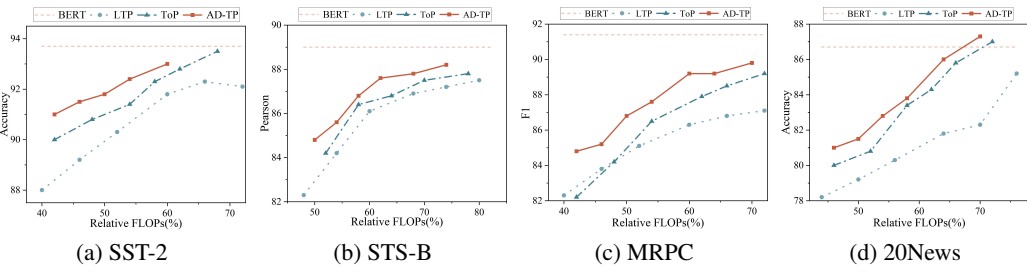

(a) SST-2     (b) STS-B     (c) MRPC     (d) 20News

Figure 8: Performance comparison of AD-TP, LTP, and ToP under different FLOPs budgets.

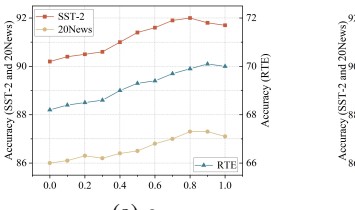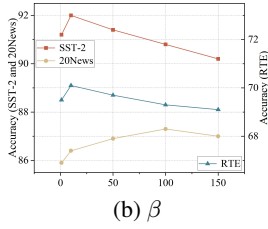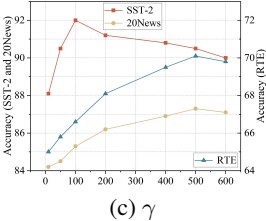

Figure 9: Impact of $\alpha$, $\beta$, and $\gamma$ on model performance.

## 5.4 Impact of $\alpha$, $\beta$, and $\gamma$

To analyze the impact of individual loss weights in the objective function, we systematically vary the hyperparameters $\alpha$, $\beta$, and $\gamma$ on the SST-2, RTE, and 20News. As shown in Fig. 9, a smaller value of $\alpha$ reduces the guidance effect from the teacher model predictions, leading to a decline in performance. Increasing $\beta$ and $\gamma$ enhances learning by improving attribution alignment and token retention. However, excessively large values may cause the model to overfit certain sub-objectives, thereby degrading overall accuracy. It is noteworthy that the optimal value of $\alpha$ remains relatively consistent across different tasks, while the best-performing values of $\beta$ and $\gamma$ vary depending on the task, indicating that attribution and retention mechanisms require task-specific tuning.

## 5.5 Impact of Candidate List Length $L_\rho$

In the process of adaptive retention ratio prediction, we define a candidate list vector $L_\rho$ containing possible retention ratios. To investigate how the length of this candidate list impacts model performance and convergence speed, we conducted comparative experiments using 3 candidate lists of different lengths. Specifically, List ① contains 5 coarse-grained values, List ② includes 10 medium-grained values, and List ③ comprises 20 fine-grained values. As shown in Fig. 10, the results demonstrate that shorter lists (List ①) enable faster convergence but achieve lower accuracy, while longer lists (List ③) result in higher accuracy but slower convergence. Considering both performance and training efficiency, we select List ② as the final configura-

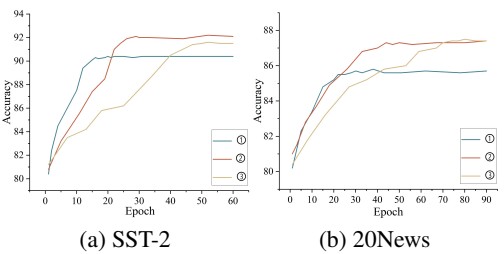

Figure 10: Impact of candidate list length on model accuracy and convergence speed in ARP.

tion, as it strikes an optimal balance between accuracy and convergence speed. Detailed configurations of the candidate list and supporting results in additional datasets are documented in AppendixA.5.

## 5.6 Evaluation of TSP

To evaluate the effectiveness of TSP, we randomly sampled 1,000 examples from the validation sets of SST-2 and MRPC, and conducted a quantitative comparison using two metrics: (i) Spearman $\rho$: measures the consistency of token-importance rankings between TSP and IG; (ii) MSE (Mean Squared Error): measures the numerical similarity between the importance scores predicted by TSP and those derived from IG. As shown in Table 5, TSP achieves high ranking consistency ($\rho > 0.75$) and low numerical error

Table 5: Consistency Between TSP and IG Scores.

| Dataset | Spearman $\rho$ | MSE |
|---------|-----------------|--------|
| SST-2   | 0.812           | 0.0119 |
| MRPC    | 0.778           | 0.0143 |

compared with IG, indicating that the module can effectively learn and approximate the saliency distribution derived from IG.

## 5.7 Sensitivity of Model Performance to the Number of IG Steps

To reduce computational overhead, we set the IG step count $m = 1$ by default in our implementation. To examine the sensitivity of model performance to this approximation, we conducted additional experiments with different values of $m \in \{1, 3, 5, 7\}$. The results are summarized in Table 6.

Table 6: Performance of AD-TP with different IG step counts.

| Dataset | m = 1 | m = 3 | m = 5 | m = 7 |
|---------|-------|-------|-------|-------|
| SST-2   | 92.0  | 92.1  | 92.5  | 92.3  |
| MRPC    | 89.8  | 89.6  | 90.0  | 90.5  |

Experimental results show that the model exhibits low sensitivity to the choice of $m$. Increasing the number of IG steps brings only slight and unstable performance improvements (typically less than 1%), while the computational cost grows approximately linearly with $m$. For example, when $m = 10$, the training time increases by about tenfold, which is impractical in resource-constrained or distributed environments. Therefore, $m = 1$ achieves a good balance between performance and efficiency.

## 6  Conclusion

In this work, we propose an attribution-driven adaptive token pruning method, AD-TP, for Transformer model compression to reduce computational resource requirements. In contrast to traditional methods that rely on accumulating attention weights to assess token importance, AD-TP applies IG to more accurately evaluate the contribution of each token to the model prediction. Furthermore, AD-TP addresses the issue of mismatched pruning ratios and sequence complexity in conventional token pruning methods. It introduces a lightweight adaptive token retainer that dynamically selects an appropriate pruning ratio based on sequence complexity to better retain important tokens. To the best of our knowledge, AD-TP is the first Transformer compression method that explicitly considers sequence complexity and implements adaptive token pruning. By combining teacher supervision with self-supervision, AD-TP effectively reduces computational overhead while maintaining performance. Extensive experimental results demonstrate that AD-TP achieves a 4% reduction in FLOPs and an 8% improvement in performance compared to existing state-of-the-art token pruning methods, showing significant advantages. **Limitations:** this work focuses only on pruning at the input layer, and extending adaptive token pruning to other layers of the model remains an important direction for future research.

## Acknowledgments

This work was supported in part by Natural Science Foundation of Shandong Province (No. ZR2022MF328 and No. ZR2019LZH014), in part by National Natural Science Foundation of China (No. 61602284 and No. 61602285), in part by State Key Lab of Processors Open Fund Project (No. CLQ202409), and in part by CCF-Ricore Education Fund (No. CCF-Ricore OF 2024003).

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

# A  Appendix

## A.1  Transformer and Token Pruning

Each basic Transformer encoder layer consists of a multi-head attention (MHA) mechanism and a feed-forward neural network (FNN) module, each surrounded by residual connections. Suppose that each Transformer layer contains $N_h$ attention heads. Given an input sequence $X = \{x_1, \ldots, x_n\}$ of length $n$, the MHA computes the importance of each token with respect to all other tokens as follows:

$$MHA(X) = \sum_{h=1}^{N_h} Att\left(W_Q^h, W_K^h, W_V^h, W_O^h, X\right) \tag{16}$$

where $W_Q^{(l,h)}, W_K^{(l,h)}, W_V^{(l,h)}, W_O^{(l,h)} \in R^{d \times d_h}$ denote the projection matrices for query, key, value, and output, respectively. Let $d$ be the hidden dimension, and $d_h$ the output dimension of each attention head, typically set to $d/N_h$. The computation for a single attention head is given by:

$$Att(W_Q, W_K, W_V, W_O, X) = W_O \cdot softmax\left(\frac{XW_Q(XW_K)^T}{\sqrt{d}}\right) \cdot XW_V \tag{17}$$

Subsequently, the output of the MHA is passed through a residual connection followed by layer normalization, as defined by the following equation:

$$X_{MHA} = LN(MHA(X) + X) \tag{18}$$

Then, $X_{MHA}$ is fed into the FNN, sequentially passes through two feed-forward layers, and is then combined with the original input via a residual connection to produce the following output:

$$FNN(X_{MHA}) = max(0, X_{MHA}W_1 + b_1)W_2 + b_2 \tag{19}$$

$$X_{out} = LN(FNN(X_{MHA}) + X_{MHA}) \tag{20}$$

where, $W_1$ and $b_1$, and $W_2$ and $b_2$, are the weights and biases of the two feed-forward layers, respectively.

To reduce the computational overhead of Transformers, token pruning removes redundant tokens from the sequence based on importance estimation. A common approach is to introduce a sparse binary mask to identify and discard unimportant tokens, thereby reducing the number of tokens involved in attention and feed-forward computations. With such a mask, the input can be reformulated as:

$$X \to X \cdot Mask_{token} \to X_{Mask} \tag{21}$$

where $Mask_{token} \in \{0,1\}^n$ indicates whether each token is retained. The outputs of the MHA and FNN modules are given by:

$$MHA_m(X) = \sum_{h=1}^{N_h} Att\left(W_Q^h, W_K^h, W_V^h, W_O^h, X_{Mask}\right) \tag{22}$$

$$X_{Mask}^{MHA} = LN(MHA_m(X) + X_{Mask}) \tag{23}$$

$$FNN\left(X_{Mask}^{MHA}\right) = max\left(0, X_{Mask}^{MHA}W_1 + b_1\right)W_2 + b_2 \tag{24}$$

The original computational complexity of the Transformer is $O(n^2)$. By introducing a sparse mask, the number of tokens involved in computation can be significantly reduced in tasks, thereby improving overall inference efficiency.

## A.2  Integrated Gradients

IG is a widely used attribution method for interpreting predictions made by neural networks. It aims to quantify the contribution of each input feature to the model output by constructing an interpolation path between the original input and a predefined baseline input, and then integrating the gradients along this path. Compared to conventional gradient-based approaches, IG effectively mitigates the gradient saturation problem and provides more stable attribution results, especially in deep networks that use nonlinear activation functions.

Given a model prediction function $F$, an input $x$, and a baseline input $x'$, the IG of the $i$-th feature is defined as:

$$IG_i(x) = (x_i - x_i') \times \int_{\alpha=0}^{1} \frac{\partial F\left(x' + \frac{k}{m}(x - x')\right)}{\partial x_i} \tag{25}$$

Since the integral is difficult to compute analytically in practice, it is typically approximated using a Riemann sum. By dividing the interval $[0, 1]$ into $m$ sub-intervals, we obtain:

$$IG_i(x) \approx (x_i - x_i') \cdot \frac{1}{m} \sum_{k=1}^{m} \frac{\partial F\left(x' + \frac{k}{m}(x - x')\right)}{\partial x_i} \tag{26}$$

Table 7: Parameter settings for different datasets.

| Dataset | Learning Rate | Epochs | Batch Size | $\alpha$ | $\beta$ | $\gamma$ |
|---------|---------------|--------|------------|----------|---------|----------|
| CoLA | 2e-5 | 20 | 32 | 0.9 | 50 | 10 |
| MNLI | 3e-5 | 30 | 16 | 0.8 | 50 | 500 |
| SST-2 | 2e-5 | 30 | 32 | 0.8 | 10 | 100 |
| QNLI | 2e-5 | 30 | 32 | 0.9 | 100 | 100 |
| MRPC | 3e-5 | 30 | 16 | 0.9 | 100 | 100 |
| QQP | 4e-5 | 30 | 16 | 1.0 | 100 | 500 |
| RTE | 2e-5 | 30 | 16 | 0.9 | 10 | 500 |
| STS-B | 5e-5 | 10 | 16 | 0.8 | 10 | 100 |

This method obtains relatively accurate feature attributions through multiple forward and backward passes, without modifying the original model architecture. In our approach, IG is used to quantify the saliency of each token. These saliency scores are then employed to estimate the importance of tokens to the model output, guiding the subsequent token retainer strategy and revealing the primary focus of the model.

## A.3 Implementation Details

All experiments are implemented using the PyTorch framework and the Huggingface Transformers library on a single NVIDIA RTX 3060 GPU. The teacher model is a fine-tuned BERT-base with 12 Transformer layers, 768-dimensional hidden units, and 12 attention heads. The student model adopts two configurations with 6 or 12 Transformer layers, keeping other hyperparameters consistent with the teacher. [2] The teacher is trained for 5 epochs on each task, and the best validation checkpoint is used for evaluation. The learning rate is fixed at 2e-5, with a batch size of 32.

For the student model, learning rates are searched in {2e-5, 3e-5, 4e-5, 5e-5, 6e-5}, and the distillation weight $\alpha$ is selected from {0.6, 0.7, 0.8, 0.9, 1.0}. Other hyperparameters $\beta$ and $\gamma$ are tuned over {1, 10, 50, 100} and {10, 50, 100, 200, 400, 500}, respectively. The candidate list for pruning ratios is defined as $L_p = \{0.1, 0.2, \ldots, 1.0\}$. Detailed task-specific configurations are reported in Table 7.

## A.4 Additional Results on ARP Effectiveness

To further support the analysis in Section 5.4, we provide additional results on the RTE and MRPC datasets. As in the main experiments, we compare three pruning strategies: Static, Random, and Adaptive (AD-TP), using the same model architecture. These approaches differ only in the method used to determine the token retention ratio. The corresponding experimental results are summarized in Table 8.

Table 8: Ablation study of the adaptive token retainer.

| Model | RTE | | MRPC | |
|-------|-----|----------|------|----------|
| | Acc. | $\rho$ | F1 | $\rho$ |
| Static | 66.2 | 0.30 | 87.4 | 0.30 |
| Random | 67.5 | 0.30 | 88.1 | 0.30 |
| Adaptive | **70.1** | 0.43 | **89.8** | 0.32 |

## A.5 Candidate List Configuration and Full Results

To support the analysis in Section 5.5, we provide the detailed configuration of the candidate list vector $L_\rho$ along with the corresponding experimental results. We define three candidate lists with varying granularity levels for adaptive retention ratio prediction:

- List ①: {0.2,0.4,0.6,0.8,1.0}

- List ②: {0.1,0.2,0.3,0.4,0.5,0.6,0.7,0.8,0.9,1.0}

- List ③: {0.05,0.1,0.15,0.2,0.25,0.3,0.35,0.4,0.45, 0.5,0.55,0.6,0.65,0.7,0.75,0.8,0.85,0.9,0.95,1.0}

The experimental results on the MRPC dataset are shown in Fig 11.

---

[2]We use the pretrained models and tokenizers from https://huggingface.co/.

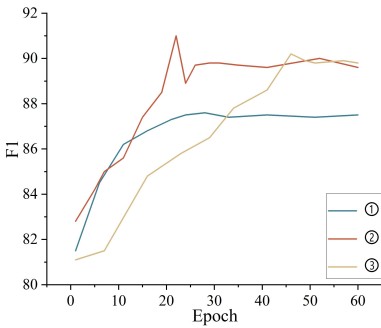

Figure 11: Impact of Candidate List Length on Model Accuracy and Convergence on MRPC.

## A.6 Inference Overhead vs. Latency

To analyze the inference overhead introduced by the Retainer module (TSP + ARP), we measured the computational cost and latency across different datasets and input lengths. The Retainer module consists of lightweight 1–2 layer feed-forward networks with fewer than 0.8M parameters, which is approximately 1% of BERT-base. Therefore, the additional inference cost is negligible compared to the Transformer backbone. Although the pruning benefits become less pronounced for shorter input sequences, the overall inference time is still reduced, as shown in Table 9.

Preliminary wall-clock latency measurements were conducted on an RTX 3060 GPU. Despite the additional modules, AD-TP achieves 30–55% reduction in end-to-end latency compared with BERT-base on SST-2, MRPC, and SQuAD, particularly for inputs longer than 64 tokens.

Table 9: Inference latency comparison between BERT-base and AD-TP on an RTX 3060 GPU.

| Dataset | Input Length | BERT-base (ms) | AD-TP (ms) | ↓ Gain |
|---------|-------------|----------------|------------|--------|
| SST-2 | 128 | 9.7 | 4.5 | 51.5% |
| MRPC | 64 | 6.1 | 3.8 | 37.7% |
| SQuADv2.0 | 512 | 21.5 | 12.3 | 42.8% |

These results demonstrate that the Retainer module introduces minimal inference overhead while providing significant end-to-end latency reduction. Future work will include evaluating AD-TP on mobile and edge CPUs to further validate its deployment efficiency in real-world settings.

## A.7 Training Cost Analysis

To analyze the additional computational cost introduced by the teacher model and Integrated Gradients (IG) supervision, we compared the training time of AD-TP and BERT-base across the eight GLUE tasks. The teacher model and IG are used only during the training phase; during inference, the proposed method relies solely on the lightweight TSP and ARP modules, requiring neither backpropagation nor teacher guidance. Therefore, the inference efficiency and deployability of the model are not affected.

As shown in Table 10, training with IG supervision increases the overall training time by approximately 20–30% compared to BERT-base training without the teacher model and IG. This additional cost is considered reasonable, given that AD-TP substantially reduces FLOPs and improves computational efficiency during inference.

These results indicate that the training-time overhead introduced by IG and the teacher model remains moderate, while offering significant improvements in inference efficiency.

## A.8 Broader Impact

This work proposes an attribution-driven adaptive token pruning method, AD-TP, for Transformer model compression to reduce computational resource consumption. The potential positive societal impacts include reduced computational and energy consumption during inference, which may contribute to more environmentally sustainable Transformer deployments. Additionally, lowering the resource requirements of widely used models like Transformer may improve accessibility for academic or industrial groups with limited computational infrastructure.

Table 10: Training time comparison between AD-TP and BERT-base on the GLUE benchmark (RTX 3060 GPU).

| Dataset | Epochs | Batch Size | BERT-base (h) | AD-TP (h) |
|---------|--------|------------|---------------|-----------|
| CoLA | 20 | 32 | 0.83 | 1.17 |
| MNLI | 30 | 16 | 33.00 | 40.50 |
| SST-2 | 30 | 32 | 3.50 | 4.33 |
| QNLI | 30 | 32 | 7.08 | 8.67 |
| MRPC | 30 | 16 | 0.25 | 0.33 |
| QQP | 30 | 16 | 22.67 | 28.83 |
| RTE | 30 | 16 | 0.17 | 0.20 |
| STS-B | 10 | 16 | 0.20 | 0.25 |

As a general-purpose model compression technique, the proposed method does not target any specific downstream application. However, there is a potential indirect risk that efficiency improvements could lower the barrier to deploying Transformer models in sensitive or harmful contexts (e.g., misinformation systems or large-scale surveillance), by making model execution cheaper. While our work is purely algorithmic and does not involve model deployment, we acknowledge this possibility.

To mitigate such risks, we encourage responsible use of model compression techniques, especially when applied to models used in real-world decision-making or content generation tasks. Transparency about pruning configurations and performance trade-offs is also essential when sharing compressed models.

