# OpenReview forum: "Attribution-Driven Adaptive Token Pruning for Transformers"
_NeurIPS.cc/2025/Conference — NeurIPS 2025 poster_

### Official Review · Reviewer_tGep · 2025-06-25

**Clarity:** 3
**Significance:** 3
**Originality:** 3
**Rating:** 5
**Confidence:** 3

**Summary:**

This paper introduces AD-TP, a novel method for token pruning in Transformer models designed to enhance computational efficiency. The authors argue that existing methods are suboptimal because they often rely on attention scores to gauge token importance, which may not correlate well with a token's actual contribution, and they use static pruning ratios that don't adapt to the varying complexity of different input sequences. To address this, AD-TP makes two main contributions. First, it uses Integrated Gradients (IG), an attribution method, to obtain a more accurate measure of each token's importance to the final prediction. Second, it introduces a lightweight "Adaptive Token Retainer" module that dynamically predicts an appropriate token retention ratio for each individual input sequence. This retainer is trained using a combination of teacher-student knowledge distillation (on both logits and IG-based attribution scores) and a self-supervision objective that is gradually phased in during training. Experiments on GLUE, SQUAD, and 20News datasets show that AD-TP significantly reduces computational costs (measured in FLOPs) while maintaining or even improving model accuracy, outperforming existing state-of-the-art token pruning techniques.

**Questions:**

Please see the weaknesses described above.

**Ethical Concerns:**

["NO or VERY MINOR ethics concerns only"]

**Final Justification:**

My final justification stays the same as above original review.

**Limitations:**

yes

**Paper Formatting Concerns:**

no concerns

**Quality:**

3

**Strengths And Weaknesses:**

Strengths:

The paper's core ideas (use of Integrated Gradients (IG) for supervising token importance) are novel and well-justified. The concept of an adaptive pruning ratio, dynamically determined for each sequence, is a key strength. The paper effectively argues that a one-size-fits-all pruning strategy is inefficient, given the wide variance in sequence length and complexity.

The architecture of the Adaptive Token Retainer, split into a ratio predictor (ARP) and a saliency predictor (TSP), is intuitive and addresses the "how many" and "which" questions of pruning separately.

The paper presents extensive experiments that demonstrate the effectiveness of AD-TP.

The paper is very well-written and organized.

Weaknesses:

The proposed method uses a teacher model and computes Integrated Gradients during training. This process is likely to be more computationally expensive and time-consuming than training methods that rely on attention scores, which are obtained almost for free during the forward pass. A discussion or quantitative analysis of this training-time overhead would provide a more complete picture of the method's trade-offs.

As the authors acknowledge in the limitations, pruning is only applied to the input embeddings. This means the quadratic complexity of self-attention is applied to the reduced sequence in all subsequent layers. While this is a valid and effective approach, it misses the potential for even greater savings from layer-wise pruning.

---

> ### Author Rebuttal · Authors · 2025-07-30
>
> We sincerely thank the reviewer for the encouraging and detailed feedback, especially the recognition of our method's **“novel and well-justified ideas”**, the **“intuitive architecture”** of ARP and TSP, and the **“extensive experiments”**. We also appreciate the comment that the paper is **“very well-written and organized”**. For each concern raised, we provide specific responses below.
>
> ***W1: Regarding Training Cost***
>
> Thank you very much for this insightful and constructive comment. Introducing a teacher model and computing IG during training does introduce additional computational cost compared to methods based solely on attention scores, which are available "for free" during the forward pass. This is indeed a meaningful trade-off that deserves further discussion.
>
> We would like to clarify the following points:
>
> 1. The teacher model and IG are used only during the **training phase**; during inference, our method relies entirely on the lightweight **TSP** and **ARP**, with no need for backpropagation or teacher guidance, which does **not affect inference efficiency and deployment**.
>
> 2. Following your suggestion, we evaluated the training time for the eight GLUE tasks on an **NVIDIA RTX 3060 GPU**. As shown in **Table R1**, compared to BERT-base training without IG and the teacher model, the overall training time increases by about **20–30%**. We believe this is a reasonable cost, especially considering that AD-TP significantly reduces FLOPs and improves computational efficiency during inference.
>
> **Table R1: Training Time Comparison between AD-TP and BERT-base on GLUE**
>
> | Dataset | Epochs | Batch Size | BERT-base Time (h) | AD-TP Time (h) |
> |---------|--------|-------------|---------------------|----------------|
> | CoLA    | 20     | 32          | 0.83                | 1.17           |
> | MNLI    | 30     | 16          | 33.00               | 40.50          |
> | SST-2   | 30     | 32          | 3.50                | 4.33           |
> | QNLI    | 30     | 32          | 7.08                | 8.67           |
> | MRPC    | 30     | 16          | 0.25                | 0.33           |
> | QQP     | 30     | 16          | 22.67               | 28.83          |
> | RTE     | 30     | 16          | 0.17                | 0.20           |
> | STS-B   | 10     | 16          | 0.20                | 0.25           |
>
> 3. Unlike attention scores, which can be affected by structural or positional biases, **IG provides attribution signals more directly tied to model predictions**, making it a stronger basis for saliency labeling. To validate this, we conducted an **ablation study** by replacing IG with attention-based heuristics. As shown in **Table R2**, IG supervision leads to higher saliency prediction accuracy.
>
> **Table R2: Comparison of the Impact of IG and Attention on Model Performance**
>
> | Dataset | Attention | IG   |
> |---------|-----------|------|
> | SST-2   | 90.2      | 92.0 |
> | MRPC    | 87.6      | 89.8 |
>
> ***W2: Regarding the limitations***
>
> We fully understand and agree with your concern regarding this limitation.
>
> In this work, we chose to apply pruning only at the input layer, primarily for the sake of **stability and controllability**. Pruning in intermediate layers often requires cross-layer coordination, which can lead to training **instability** or **interfere with semantic representations**.
>
> Nevertheless, our method still achieves **significant inference acceleration** and **competitive accuracy** on multiple standard benchmark datasets (see Tables 1, 2, and 3), indicating that the TSP and ARP modules are **effective** under the current setting.
>
> Moreover, our framework is **modular** by design, and the TSP module can be naturally **extended** to support token pruning in intermediate layers. We are actively exploring this direction and plan to **incorporate cross-layer pruning strategies** in future work to further improve efficiency.
>
> We believe this is a promising direction for further research, and we sincerely appreciate your valuable suggestion.

---

### Official Review · Reviewer_QJNG · 2025-07-01

**Clarity:** 4
**Significance:** 3
**Originality:** 4
**Rating:** 5
**Confidence:** 4

**Summary:**

This paper introduces AD-TP (Attribution-Driven Adaptive Token Pruning), a novel method for compressing Transformer models by dynamically pruning less informative tokens from input sequences. The work aims to address two key limitations of prior token pruning techniques: (1) the use of inaccurate proxies like attention scores for determining token importance, and (2) the reliance on static, one-size-fits-all pruning configurations. To achieve this, AD-TP makes two main contributions. First, it employs Integrated Gradients (IG), a feature attribution method, to compute a more faithful measure of each token's contribution to the final prediction. Second, it introduces a lightweight, Attribution-Aware Adaptive Token Retainer module that operates at the instance level. This module predicts an optimal token retention ratio for each specific input sequence and learns to approximate the expensive IG scores for efficient inference. The entire system is trained via a teacher-student distillation framework, where a larger teacher model provides supervision for both the classification task and the IG-based attribution scores.

**Questions:**

1.  Federated Implementation Feasibility: How do the authors envision AD-TP being trained in a federated environment?
- Would this be a "Federated Distillation" setup where a central server holds the teacher model and clients only train the student and retainer?
- If the entire process is federated, how would the expensive IG calculations be handled on resource-constrained client devices? What would be the communication cost of aggregating the adaptive retainer module in each round?

2.  Generalization to Other Architectures: The ARP module relies on a `[CLS]` token. Have you considered how to adapt this mechanism for decoder-only models like the GPT family? Would a simple approach like averaging final-layer token embeddings suffice as a global sequence representation?

3.  Sensitivity to IG Approximation: The IG calculation is approximated using `m=1` step for efficiency. How sensitive is the final model performance to this approximation? Is there a significant accuracy gain from using a larger `m`, and how does that affect the already high training cost?

**Ethical Concerns:**

["NO or VERY MINOR ethics concerns only"]

**Final Justification:**

Authors have conducted additional experiments for Q3 & W2 to address my concerns. My concerns have been clarified to a great extent, and I am satisfied with the response. I am keeping my original score.

**Limitations:**

Yes

**Quality:**

3

**Strengths And Weaknesses:**

Strengths:

- Quality & Originality: Principled Saliency Metric - The core innovation of using Integrated Gradients (IG) instead of attention scores as the basis for pruning is a significant and well-justified step forward. The authors correctly argue that attention weights are not designed to measure a token's contribution to the final output. By grounding the pruning decision in a more theoretically sound attribution method like IG, the method directly links pruning to the model's task-specific reasoning. This is a novel and principled approach that, as the results show, preserves accuracy more effectively.

- Significance & Originality: Instance-Level Adaptability - The introduction of an adaptive module that predicts the pruning ratio for each input is a crucial advantage. This is particularly relevant in heterogeneous data environments, such as those found in Federated Learning (FL). In an FL setting, client data is often non-IID; some clients may have simple, highly prunable data, while others have complex data requiring more tokens. AD-TP's ability to adapt its compression rate on-the-fly for each sample is a major improvement over static methods that would be suboptimal across a diverse federation of clients.

- Quality: Comprehensive Training Framework - The teacher-student training regime is methodologically robust. The combination of logits-based distillation and attribution-based distillation provides a rich supervisory signal, ensuring the student model learns both *what* to predict and *why* it is making that prediction. The scheduled weighting of the attribution loss is a clever detail that likely aids training stability.

- Significance: Strong Empirical Results - The paper demonstrates impressive performance, substantially reducing FLOPs while maintaining high accuracy across a range of NLP tasks. The fact that AD-TP outperforms other well-established compression techniques like DistilBERT and specialized pruning methods like ToP underscores its effectiveness, especially on long-sequence tasks where the computational savings are most impactful.

Weaknesses:

- Quality & Significance: Prohibitive Training Complexity - The most significant weakness of AD-TP, particularly from a Federated Learning perspective, is its computationally expensive training process. The methodology requires a pre-trained teacher model, the calculation of IG scores (which involves multiple forward/backward passes per sample), and the optimization of a complex, multi-part loss function. This high resource requirement makes the method impractical for on-device training in a conventional FL setting. It appears better suited for centralized training followed by deployment to the edge, rather than end-to-end federated training.

- Quality: Inference Overhead vs. Latency - While the paper focuses on FLOPs reduction, the Adaptive Token Retainer module itself adds parameters and a non-zero computational overhead at inference time. For very short sequences, this overhead could potentially negate the benefits of pruning. A more complete evaluation would include wall-clock latency benchmarks on hardware relevant to edge deployments (e.g., mobile CPUs) to demonstrate a practical speedup beyond theoretical FLOPs.

- Clarity & Significance: Unexplored Federated Learning Implications - While the adaptability of AD-TP is a great fit for FL's data heterogeneity, the paper does not discuss how the method would be implemented in such a setting. It is unclear how the complex training would be federated, or what the communication overhead of aggregating the adaptive retainer module across clients would be. Without this analysis, its applicability to the FL domain remains speculative.

- Originality: Architectural Specificity - The design of the Adaptive Retention Ratio Predictor (ARP) relies on the `[CLS]` token embedding as a global representation of the input. This is standard for BERT-like encoders but limits the method's direct applicability to other popular architectures, such as decoder-only models (e.g., GPT-family) or encoder-decoder models (e.g., T5), without non-trivial modifications.

- Syntactic errors:
1. Figure 4 Caption: "Adaptive Aoken Aetainer" -> "Adaptive Token Retainer"

---

> ### Author Rebuttal · Authors · 2025-07-27
>
> We sincerely thank you for your detailed comments and constructive feedback, especially your appreciation of our work as a **"novel and principled approach"**, **"methodologically robust"**, **"major improvement"**, **"clever detail"**, **"training stability"**, **"impressive performance"**, and **"significant and well-justified step forward"**. Your thoughtful suggestions greatly help us improve the clarity and completeness of the paper.  We provide detailed responses to your comments below.
>
> ***Q1: Feasibility and Design of AD-TP in Federated Learning Environments***
>
> We sincerely thank the reviewer for raising this important and practically meaningful question. We fully agree that the proposed **Federated Distillation** framework is a feasible and effective way to deploy AD-TP.
>
> In this framework, a central server holds the teacher model and performs attribution analysis on a public or synthetic dataset, **precomputing supervision signals** such as logits and IG (Integrated Gradients) saliency scores. Clients use their **local private data** to train only the **lightweight student model and the Retainer modules (ARP / TSP)**. Each client receives the supervision signals (e.g., class-level logits and attribution distributions) from the server and uses them for local training. This design ensures **data privacy**, shifts **heavy computation** to the server, and supports **client heterogeneity**.
>
> **Client-side IG Computation**
>
> Computing IG on the client side is **optional**. By setting $k = 0$ in Equation (6), the student model can fully rely on attribution scores provided by the teacher model, with **no need to compute IG locally**.  If the goal is to improve the student’s attribution ability via self-supervision, a **lightweight variant** such as IG with \( m = 1 \) can be used.  For instance, with a sequence length of 128, a single IG computation with \( m = 1 \) requires only **3.9 GFLOPs**, which is negligible on modern GPUs and therefore **practically feasible**.
>
> **Communication Cost of the Retainer**
>
> The Retainer in AD-TP consists of two lightweight modules, **ARP and TSP**, totaling approximately **0.8M parameters** (about **1% of BERT-base**).  Under **32-bit precision**, each communication round (**upload + download**) incurs a cost of about **6.4MB**, which is acceptable for most federated learning scenarios.  Furthermore, techniques such as 8-bit quantization or Top-k sparse updates can be applied to further reduce communication overhead, making the approach viable even in low-bandwidth federated environments.
>
> ***Q2: Adapting ARP for Architectures Without a [CLS] Token (e.g., GPT)***
>
> Thank you for this great question. We have indeed considered adapting the ARP module to decoder-only architectures (such as the GPT family), particularly in the absence of a `[CLS]` token.
>
> Regarding your suggestion of averaging the final-layer token embeddings, we agree that this is a reasonable and effective simplification.
>
> In addition, we have explored two alternative mechanisms:
>
> - **Attention Pooling**: We introduce a lightweight attention layer that assigns a learnable weight $t_i$ to each token embedding $e_i$, constructing a global representation as
>   $$
>   x_g = \sum t_i \cdot e_i.
>   $$
>   This approach serves as a soft aggregation strategy, independent of any special structural token, and offers strong global awareness and task adaptability.
>
> - **Last Token Representation**: In generative tasks, the final token in a sequence often carries rich semantic information. We therefore also consider using the last token embedding as a simple and practical global representation.
>
> Once again, thank you for raising this important question, which helped us deepen our reflection on the generalizability of the proposed method.
>
> ***Q3: Sensitivity of Model Performance to IG Approximation***
>
> Thank you for raising this thoughtful question.
>
> In our implementation, we set *m = 1* by default to reduce the computational burden. To evaluate the sensitivity of the model to this approximation, we conducted additional experiments with different step values (*m* = 1, 3, 5, 7). The results are shown in Table R1 below:
>
> **Table R1: Performance of AD-TP with different IG step counts**
>
> | Dataset | m = 1 | m = 3 | m = 5 | m = 7 |
> |---------|-------|-------|-------|-------|
> | SST-2   | 92.0  | 92.1  | 92.5  | 92.3  |
> | MRPC    | 89.8  | 89.6  | 90.0  | 90.5  |
>
> **Key Findings:**
>
> - The model exhibits **low sensitivity** to the choice of *m*: increasing *m* results in only minor performance improvements (typically less than 1%).
> - These gains are **inconsistent across tasks**, and in some cases, performance even slightly decreases due to noise introduced by approximation.
>
> **Cost Analysis:**
>
> - The computational cost of IG grows **linearly** with *m*.
> - For example, setting *m = 10* increases the training cost by approximately **10×**, which is particularly prohibitive in **resource-constrained or distributed environments** (e.g., federated learning).
>
> We believe that *m = 1* strikes an optimal balance between **performance and efficiency**, making it a practical and scalable choice in real-world applications.
>
> ***W1: Prohibitive Training Complexity***
>
> Thank you for raising this important concern. We fully acknowledge that AD-TP introduces additional overhead during training compared to conventional pruning methods, primarily due to the use of IG (Integrated Gradients) attribution and the teacher–student framework.
>
> However, we would like to clarify several points to demonstrate that AD-TP remains feasible in practical settings:
>
> - **IG computation and teacher supervision are only used during training**, while inference remains **efficient and architecture-agnostic**;
> - We adopt an **approximate IG method with $m = 1$** , which significantly reduces computational cost with minimal performance loss;
> - The Retainer modules (ARP and TSP) are extremely **lightweight**, with a total parameter count of approximately **0.8M**.
>
> In federated learning scenarios, we can adopt a **Federated Distillation** strategy: the teacher model is hosted on a central server, which computes **logits and attribution scores** on public or synthetic data. Clients only need to receive these supervision signals to train the **student model and Retainer modules**. For implementation details, please refer to our response in **Q1**.
>
> While AD-TP is compatible with **centralized training followed by deployment to edge devices**, it is **not limited** to this setup. Its **modular design** also makes it applicable to **end-to-end federated training**, which we plan to explore further in future work.
>
> ***W2: Inference Overhead vs. Latency***
>
> Thank you for raising this important question. We understand the concerns regarding inference overhead and offer the following clarifications.
>
> The Retainer module (TSP + ARP) consists of small 1-2 layer feed-forward networks with **fewer than 0.8M parameters** (about 1% of BERT-base). The added inference cost is **negligible** compared to the Transformer backbone. However, it is true that the pruning benefits may be smaller for shorter sequences. Nonetheless, the inference time is still reduced, and even for short sequences, there is **some benefit**, as shown in Table R2.
>
> We appreciate the reviewer’s suggestion regarding wall-clock latency benchmarks. As our focus has been on algorithm design and FLOPs reduction, deployment on real edge devices (e.g., mobile CPUs) is still in progress. We have conducted preliminary tests on an RTX 3060 GPU. As shown in Table R2, despite the added modules, AD-TP reduces end-to-end latency by **30–55%** compared to BERT-base on SST-2, MRPC, and SQuAD, especially for inputs ≥64 tokens.
>
> **Table R2: Inference Latency on RTX 3060**
>
> | Dataset   | Input Length | BERT-base (ms) | AD-TP (ms) | ↓ Gain   |
> |-----------|---------------|----------------|------------|----------|
> | SST-2     | 128           | 9.7            | 4.5        | 51.5%    |
> | MRPC      | 64            | 6.1            | 3.8        | 37.7%    |
> | SQuADv2.0 | 512           | 21.5           | 12.3       | 42.8%    |
>
> We agree that mobile/edge deployment is important and plan to evaluate AD-TP on mobile CPUs in future work.
>
> ***W3: Unexplored Federated Learning Implications***
>
> We thank the reviewer for the insightful comment on the applicability of AD-TP in federated learning (FL). While this paper primarily focuses on centralized training, we agree that its adaptive design aligns well with data heterogeneity in FL. Relevant strategies have been detailed in **Q1** and **W1**. We again appreciate your suggestion, which helped broaden the scope of AD-TP beyond centralized settings.
>
> ***W4: Architectural Specificity***
>
> We thank the reviewer for this insightful comment. Although ARP relies on the [CLS] token in BERT, as mentioned in **Rebuttal Q2**, it can be adapted to architectures like GPT or T5 using alternative methods such as **mean pooling, attention pooling, or using the last token**. We plan to continue evaluating the performance of these methods in future work and further optimize their applicability.
>
> ***W5:  Syntactic errors***
>
> We thank the reviewer for pointing out the typo in Figure 4's caption. “Adaptive Aoken Aetainer” will be corrected to “Adaptive Token Retainer” in the revision. We have also checked the full manuscript to ensure no similar issues remain. We appreciate your attention to detail.

---

> > ### Comment · Reviewer_QJNG · 2025-08-05
> >
> > I thank the authors for conducting additional experiments for Q3 & W2 to address my concerns. I am satisfied with the response. I am keeping my original score.

---

> > > ### Author Response · Authors · 2025-08-05
> > >
> > > We are glad that our response was able to address your concerns, and we sincerely appreciate your time and engagement in the review process.

---

### Official Review · Reviewer_vH7T · 2025-07-03

**Clarity:** 2
**Significance:** 3
**Originality:** 2
**Rating:** 4
**Confidence:** 5

**Summary:**

This paper proposes AD-TP (Attribution-Driven Adaptive Token Pruning), a method for efficiently compressing Transformer models by selectively pruning tokens based on their actual contribution to the model's prediction. Unlike prior approaches that rely on attention scores or static thresholds, AD-TP leverages Integrated Gradients (IG) to estimate token saliency more accurately and introduces a lightweight adaptive token retainer that dynamically adjusts the pruning ratio per input sequence and retain tokens based on token saliency.  Extensive experiments on GLUE, SQuAD, and 20News and more show that AD-TP significantly reduces computational cost (up to ~14× FLOPs reduction) while preserving or improving model accuracy, outperforming existing pruning and distillation methods.

**Questions:**

1. It seems like both IG and TSP affect the token saliency. Why do we have two modules to determine the token saliency, and how do they collaborate to work together?

2. Why do we use IG instead of other metrics like (masked) KL divergence? Can you show the benefit of IG over other metrics such as (masked) KL divergence?

3. In Table 1 and Table 3, why is AD-TP6 always better than BERT6 (S) although the FLOPs are much lower? It seems like this is not always true for all token pruning methods, such as CoFi6. Any insights here?

4. From Figure 2, it seems like there is a length distribution mismatch between the training set and the evaluation set. Does that have any impact on the ARP module, which is used to estimate the complexity of the sequence and determine the retention ratio? For example, there may be domain shift in terms of input length—how does this domain shift affect your method?

If my concerns about IG are properly addressed, I would consider raising my rating. Additional insights would be appreciated.

**Ethical Concerns:**

["NO or VERY MINOR ethics concerns only"]

**Final Justification:**

The rebuttal clarifies the function and contribution of each component more clearly. I raise my rating to borderline accept.

**Limitations:**

yes

**Quality:**

3

**Strengths And Weaknesses:**

**Quality**: This work aims to improve the token pruning method, and the idea of estimating sequence-level difficulty and assigning an optimal pruning ratio for each sequence is technically sound.

**Clarity**: The motivation, method, and experiments are presented completely, but there is room for improvement. For example, the tables are too small, especially Table 1. Also, Figure 3 seems useless, and I don't quite understand what it's trying to convey. Additionally, it remains unclear what is determining the token saliency—is it the TSP module or the IG?

**Significance**: How to dynamically determine the pruning ratio for each input sequence deserves further exploration by the community, and the ablation study on the impact of candidate list length seems interesting. However, it lacks an ablation on the ARP module alone, so it is hard to judge what its contribution looks like. It would be nice to see the ablation study for each individual component (IG, ARP, TSP).

**Originality**: See **Significance**. In general, some interesting insights are provided, but the analysis is not adequate.

---

> ### Author Rebuttal · Authors · 2025-07-30
>
> We sincerely thank the reviewer for the detailed and constructive feedback, as well as the recognition of our work as a **“technically sound”**, **“presented completely”** approach with **“interesting insights”** and **“good evaluation”**. We address each of your thoughtful questions and concerns in detail below.
>
> ***Q1: Role and Interaction of IG and TSP in Saliency Estimation***
>
> Thank you for raising this insightful question. Indeed, both Integrated Gradients (IG) and Token Saliency Predictor (TSP) relate to token saliency, but they play complementary roles within AD-TP:
>
> - **IG** is used to compute token-level attributions, serving as the core measure of each token’s contribution to the model’s predictions. It is used **only during training** to provide fine-grained, attribution-based token saliency scores that accurately reflect each token's **true contribution** to the model’s output. These scores are then used as **supervision** to train the TSP module.
>
> - **TSP** is a lightweight, learnable, dynamic module that, after training, efficiently approximates the IG scores **during inference**, thus avoiding the high computational cost of backpropagation-based attribution methods.
>
> The collaboration between the two allows the model to benefit from the **high-quality supervision signals** provided by IG during training, while enabling **fast estimation** via TSP during inference. TSP can also work collaboratively with the ARP module to jointly decide which tokens to retain, enabling the learning of an efficient and task-aware pruning strategy.
>
> ***Q2: Why do we use IG instead of other metrics like (masked) KL divergence? Can you show the benefit of IG over other metrics such as (masked) KL divergence?***
>
> Thank you for this valuable question. While both Integrated Gradients (IG) and KL divergence (specifically, masked KL divergence) can be used to evaluate token importance, they are based on different principles. IG is better suited to our task for the following reasons:
>
> - **Different Attribution Mechanisms**:
>   KL divergence indirectly evaluates importance by observing how the output **distribution changes** when a token is masked, making it hard to clearly attribute predictions. In contrast, IG directly integrates gradients to **quantify each token’s contribution precisely**.
>
> - **Stronger Stability and Interpretability**:
>   IG integrates gradients along the input path, producing **smoother and more robust** attributions. KL-based methods are **sensitive** to perturbations and often yield **unstable** importance scores.
>
> - **Higher Computational Efficiency**:
>   KL divergence requires masking each token and performing **a separate forward pass for each**, which is **computationally expensive**. IG enables **parallel computation** across the entire sequence, offering better efficiency.
>
> - **Better Contextual Awareness**:
>   IG captures **interactions between tokens** by considering the joint gradients of the full input. KL divergence, which focuses only on the effect of removing individual tokens, is prone to **overlooking important contextual dependencies**.
>
> ***Q3: Why is AD-TP6 better?***
>
> Thank you for this insightful question. We believe this phenomenon is primarily due to the following three factors:
>
> 1. **Task-aware adaptive token pruning**
>    AD-TP6 dynamically identifies and removes **redundant or low-information** tokens (e.g., stop words) based on saliency scores. This increases input **information density** and allows attention to focus on **task-critical content**.
>
> 2. **Better alignment between pruning objective and task needs**
>    CoFi6 is a structured pruning method that reduces FLOPs by removing layers, attention heads, or FFN units. However, at high sparsity levels, it may inadvertently prune **components crucial** for specific tasks, degrading performance. In contrast, AD-TP6 applies fine-grained, token-level pruning that better preserves semantic content and avoids structural disruption.
>
> 3. **Distillation-guided training**
>    AD-TP6 incorporates **knowledge distillation** from a teacher model, which helps the student model retain strong representational and predictive capabilities under compression.
>
> ***Q4: Impact of Input Length Distribution Shift on ARP Module***
>
> Thank you for this important question regarding potential domain shift in input length.
>
> We acknowledge that domain shift in input length between training and evaluation sets can pose challenges for fixed pruning strategies. However, AD-TP is explicitly designed to handle such variability. Our ARP module estimates the token retention ratio dynamically during inference, not based on raw sequence length, but on **semantic-level features** such as the [CLS] representation and IG-based saliency. These features capture aspects like **semantic density** and **structural complexity**, which are more indicative of task-relevant content than length alone.
>
> In fact, this design was motivated in part by the observation of length mismatches in real datasets (e.g., SST-2, shown in Figure 2). Despite this mismatch, AD-TP achieves **top performance** (Table 1), demonstrating robustness to such domain shifts. Furthermore, our method also performs consistently well on datasets with **much larger input length variation** (e.g., 20News, Table 2), further validating ARP’s generalization ability.
>
> Therefore, although input length distribution may shift, the impact on our method is minimal due to ARP’s **adaptive, complexity-aware pruning** mechanism.
>
> ***W1: Technically Sound***
>
> Thank you for your positive feedback. We appreciate your recognition of our method’s technical soundness.
>
> ***W2: Table 1 size is too small, Figure 3 seems unclear and unnecessary, and the determination of token saliency is unclear***
>
> Thank you for your constructive feedback. We greatly appreciate your comments on the presentation and clarity of our work. Below, we provide responses to each of the points you raised:
>
> 1. **Table Size**: We will enlarge Table 1 and improve its layout in the revised version to enhance readability.
>
> 2. **Figure 3**: Figure 3 is intended to show that even within the same dataset, different input sequences may have **varying structural complexity**, which can be reflected in their syntactic dependency structures. We display several example syntactic dependency graphs from the same dataset to **support the motivation** behind the ARP module design — dynamically adjusting the retention ratio based on sequence complexity.
>
> 3. **Token Saliency Determination**: The saliency during inference is determined by the TSP, which uses IG as a supervision signal during training. Please refer to the response in Q1 for further details.
>
> ***W3: Ablation on ARP module and other components***
>
> Thank you for your thoughtful and constructive feedback.
>
> Regarding the ablation of the ARP module, we have already conducted an ablation study on this component in Section 5.2 (**“Impact of the ARP”**) and Table 4 of the main text. In this study, we compare three pruning strategies: (i) fixed ratio, (ii) randomly selected ratio from a list, and (iii) our proposed adaptive ARP. The results show that ARP provides consistent performance gains, demonstrating its effectiveness.
>
> **Table 4: Ablation study of the adaptive token retainer**
> |Method|SST-2 Acc.|SST-2 ρ|20News Acc.|20News ρ|
> |-----------|------------|---------|-------------|----------|
> |Static| 91.2| 0.30| 85.7| 0.30|
> |Random| 90.5| 0.30| 82.5| 0.30|
> |Adaptive| 92.0| 0.29| 87.3| 0.27|
>
> For the ablation analysis of other components, we have already conducted ablation experiments on multiple loss terms in Section 5.2 (**"Impact of the Teacher Model and Loss Components"**) and **Figure 7** ($L_{\text{CE}}$, $L_{\text{logit}}$, $L_{\text{IG}}$, and $L_{\text{ARP}}$ + $L_{\text{TSP}}$). To further disentangle the contributions of each component, we have added results in Table R1 that ablate $L_{\text{ARP}}$ and $L_{\text{TSP}}$ individually.
>
> **Table R1: Ablation Study of $L_{\text{ARP}}$ and $L_{\text{TSP}}$ Loss Components**
>
> |Dataset|w/o $L_{\text{ARP}}$|w/o $L_{\text{TSP}}$|AD-TP|
> |---------|---------------------|---------------------|--------|
> |SST-2|86.5|86.1|92.0|
> |RTE| 61.5 | 61.3 |70.1|
> | MRPC| 86.7|86.2| 89.8|
> | 20News| 83.0 | 82.8| 87.3|
>
> In addition, we have added **structural ablation** studies on each core module, including:
> - *IG*: Replace IG with attention scores for training.
> - *ARP*: Replace adaptive pruning with random ratio.
> - *TSP*: Remove TSP, apply random pruning.
>
> As shown in Table R2, removing any single module leads to a notable performance drop. Using attention scores instead of IG fails to accurately reflect token importance, highlighting IG's **critical role** in TSP learning. Eliminating ARP results in fixed pruning ratios, which cannot adapt to input complexity and may **discard key information**. Completely removing TSP makes pruning decisions less discriminative, causing many important tokens to be **mistakenly pruned** and leading to a significant performance decline. These results underscore the essential role of each core component in the effectiveness of AD-TP.
>
> **Table R2: Structural Ablation of IG, ARP, and TSP Modules**
>
> |Dataset|Replace IG|Remove ARP|Remove TSP|AD-TP|
> |---------|------------|------------|------------|--------|
> |SST-2|90.2|80.5|79.0|92.0|
> |MRPC|87.6|80.9|78.6|89.8|
>
> ***W4: The analysis lacks depth***
>
> Thank you for your valuable feedback. We are pleased that you recognized some of the innovations in our work.
>
> To address the analysis concerns, we have added **loss and module ablation studies** (Tables R1–R2), and included an **effectiveness analysis of the TSP module** (see our response to Reviewer joVc, W4). We also introduced additional **baseline comparisons** (e.g., TS-DTP, SOA, APS).
>
> Furthermore, we plan to incorporate **visualizations** (e.g., attention maps) to better illustrate how AD-TP6 preserves key information.

---

> > ### Comment · Reviewer_vH7T · 2025-08-02
> >
> > Thank you for the detailed ablation study presented in the rebuttal. It clarifies the function and contribution of each component more clearly. I will raise my rating accordingly.

---

> > > ### Author Response · Authors · 2025-08-03
> > > **Thanks for the response**
> > >
> > > We sincerely appreciate your engagement in the review process. Thanks again for the review, and if there are any outstanding concerns that we can address, please let us know.

---

### Official Review · Reviewer_joVc · 2025-07-03

**Clarity:** 2
**Significance:** 2
**Originality:** 3
**Rating:** 4
**Confidence:** 2

**Summary:**

This paper presents a novel token pruning method for transformer-based models, which utilizes token saliencies from intergrated gradients to reduce the overall input sequence length. The proposed method, AD-TP, is composed of two distinct parts, an Adaptive Retention Ratio Predictor that predicts how much of the input sequence should be preserved (based on complexity), and a Token Saliency Predictor which predicts and selects the actual tokens to keep vs discard. The proposed method is evaluated against several existing token pruning methods, using BERT-base as the model and GLUE, SQuAD and 20News as the datasets. The results show that the AD-TP matches or outperforms the best previous method, while pruning more tokens (i.e. higher efficiency gains). The various AD-TP submodules are trained using a student-teacher framework, where the submodules learn to predict both the retention ratio and IG saliency scores without needing the ground truth.

**Questions:**

- Figure 3 is used to depict the complexity of the input sample? Can one just use a syntactic dependency parser as a light weight alternative to ARP?
- Why not just use a regression model to predict the retention ratio P_r directly instead of using a candidate list L_p?
- [1,2] are vision papers, not talking about why GPT/BERT are central to natural language processing. CoFi's citation is also [2], but it doesn't seem to be mentioned in that paper anywhere; are these references here by mistake?

Please also take a look at the questions mentioned in the *strengths and weaknesses* section.

**Ethical Concerns:**

["NO or VERY MINOR ethics concerns only"]

**Final Justification:**

The rebuttal clarifies quite a bit of the methodological concerns I had regarding the setup + efficacy of individual components. I still stand by my original assessment that the original manuscript is difficult to read, replicate and draw concrete implementations from, but the rebuttal is a good start in the right direction.

**Limitations:**

yes

**Quality:**

2

**Strengths And Weaknesses:**

The paper studies an important (and timely) problem of high inference cost in transformer-based models. The proposed methodology combines quite a few existing pieces in a novel formulation, and presents an interesting framework to train the submodules of this framework. The evaluation is quite diverse in terms of dataset, but only relies on a single (small) model. At least on this setting, the proposed methodology performs quite well, matching or improving upon the previous best method at higher efficiency gains.

Having said this, there are some weaknesses that impact the quality and relevance of this work quite significantly:

- The paper is a bit difficult to read in general, and the widespread use of similar abbreviations really makes it even harder. There are also several claims and terms used in the paper without any clear explanation or evidence of support. One instance of this is "residual strategy" in Section 3.1, which is not defined anywhere in the paper.
- The experimental setting is quite dated; the model and the datasets are really from the "previous" era of NLP, as evidenced by the fact that the related work is all 4 or 5 years old. Its really a big miss to not have tried this method on a more recent LLM model, where the impact could be much greater.
- Details on how AD-TP is trained are very sparse, apart from the loss function derivation. On what dataset is the training conducted on? How long is the training run for? This hinders reproducibility significantly, since without a trained AD-TP model the proposed methodology is incomplete.
- It was also strange to see no evaluation of TRP; do the scores predicted by it match the IG scores (after the model has been trained)? Its a fundamental piece of the overall method, and given the importance of IG scores (as self-described in the paper), its important that this module mirror IG's scores well.

Overall, the proposed methodology itself is sound, and the experimental results do improve upon existing methods to some extent, but it is unclear what the impact of the work is in the era of modern LLMs where classification is done differently, and a lot of tasks are dominated by generation (which is completely ignored in this paper). This, along with some key missing details sheds some doubt over the efficacy of the work.

---

> ### Author Rebuttal · Authors · 2025-07-30
>
> We sincerely thank the reviewer for the thoughtful feedback and the recognition of our method as **“a novel formulation”**, **“an interesting framework”**, and **“a sound methodology”** that achieves **“higher efficiency gains”** and **“performs quite well”**. For each of the concerns raised, we provide detailed responses below.
>
> ***Q1: Is Figure 3 illustrating input complexity, and can a dependency parser serve as a lightweight replacement for ARP?***
>
> Thank you for the insightful question. Yes, Figure 3 illustrates that input sequences within the same dataset can vary significantly in complexity.
>
> Using a **dependency parser** as a lightweight alternative is an interesting idea. While dependency parsers can capture **surface-level structural information**, they fail to adequately reflect **semantic factors** that are critical to model predictions, such as negation, sentiment polarity, and task difficulty. Additionally, dependency parsers are **general-purpose tools** and typically **lack task awareness**.
>
> In contrast, our ARP module is trained on the model’s own **semantic representations** (the `[CLS]` embedding), enabling it to learn **task-aware** input complexity in an end-to-end manner. This allows ARP to better align with the model’s **internal understanding** of inputs. With just a **single-layer feed-forward network**, ARP is lightweight, efficient, free of external dependencies, and easily transferable across tasks.
>
> ***Q2: Why not just use a regression model to predict the retention ratio $P_r$ directly instead of using a candidate list $L_p$?***
>
> Thank you for this insightful question. We found that directly regressing the retention ratio $P_r$ leads to **instability during training**. It is **sensitive** to initialization and label scale, and often produces **unreasonable predictions** (e.g., overly high or low retention rates), which negatively affect pruning performance. Moreover, continuous outputs **lack bounded constraints**, making it difficult to control FLOPs.
>
> In contrast, using a discrete candidate list (e.g., $L_p = \\{0.3, 0.5, 0.7, 1.0\\}$) is **more stable and controllable**. By adjusting the granularity and range of the candidate set, we can **flexibly control model FLOPs** to meet different deployment budgets and enable dynamic scheduling during inference.
>
> Therefore, we adopt the candidate list approach, which offers **bounded and interpretable predictions**, **more stable and faster training**, and **better suitability for budget-constrained applications**. In future work, we plan to explore ways to mitigate the instability and uncontrollability of regression-based prediction, making it more robust for our task and further improving overall performance.
>
> ***Q3: References Citation***
>
> We sincerely apologize for the citation issue and thank the reviewer for the careful and detailed observation.
>
> We recognize that references [1] and [2] are primarily from the computer vision domain. Although they mention the application of Transformer architectures in NLP within their related work sections, they are not the most appropriate sources to support our argument. We will replace these references in the revised version with more suitable ones, such as:
> - Dai, D., et al. (2023). **Why Can GPT Learn In-Context? Language Models Secretly Perform Gradient Descent as Meta-Optimizers**. Findings of the Association for Computational Linguistics: ACL 2023.
> - Anelli, V., et al.(2022, October). **Interpretability of BERT Latent Space through Knowledge Graphs**. In Proceedings of the 31st ACM International Conference on Information & Knowledge Management (pp. 3806–3810).
>
> We also acknowledge the incorrect citation of CoFi as reference [2]. This will be corrected in the revision and replaced with the proper original source:
> - Xia, M., et al. (2022, May). **Structured Pruning Learns Compact and Accurate Models**. In Proceedings of the 60th Annual Meeting of the Association for Computational Linguistics (Volume 1: Long Papers) (pp. 1513–1528).
>
> ***W1: Regarding the abbreviation issue***
>
> Thank you for the valuable feedback.
>
> We recognize that the paper includes an excessive use of abbreviations, particularly the repeated use of the two modules in our model: **TSP** and **ARP**. Although all abbreviations are defined upon their first appearance, we will reduce their repetition in the revised version to improve readability.
>
> Regarding the term "Residual Strategy" in Section 3.1, it refers to an **upper-bound** method that relies on label and gradient information. This method approximates a token's contribution to the model's loss by comparing hidden representations from different layers ($H_r$ and $H_l$, where $r > l$), and directly measures each token's maximum impact on the model's output. The specific formula is:
> $$
> I = \frac{\partial \text{loss}}{\partial H_r} \cdot (H_r - H_l)
> $$
> However, since it is a **post-hoc** method, it can only be computed when the labels are known and backpropagation is available. Therefore, it cannot be used in the inference phase and serves solely as a theoretical benchmark.
>
> ***W2: Regarding the experimental setting issue***
>
> Thank you for this important comment. We believe applying our method to LLMs is indeed meaningful and promising. In this work, we chose BERT-base and standard datasets (e.g., GLUE, 20News, SQuAD2.0) mainly for the purposes of **fair comparison, reproducibility, and experimental efficiency**, which are common practices in recent pruning studies (e.g., LTP, ToP, TS-DTP, APS).
>
> Encoder-only models **remain valuable** in practical latency-sensitive applications such as mobile NLP and search. In contrast, LLMs like GPT-3 or LLaMA incur much **higher training and ablation costs**, making systematic experimentation difficult. Thus, we focus on a **stable and controlled setup** as a foundation for future extensions.
>
> Our proposed ARP and TSP modules are designed to be **general and extensible**, and can be naturally applied to LLM scenarios. For example, they can be used to prune redundant tokens in large prompts while preserving task-critical elements (e.g., instructions, key phrases). We plan to explore and validate such extensions in future work.
>
> We also appreciate the reviewer’s suggestion regarding the recency of related work. We have cited multiple recent studies in our paper, including two in 2022, one strong KDD 2023 baseline, and other recent pruning and compression works. In addition, Table R1 supplements our method with a comparison against a token pruning approach that **was unpublished at the time of our submission**, as well as two other compression methods (These two methods are less closely related to our work and were therefore not discussed in detail in the main paper).
>
> **Table R1: Performance Comparison with Recent Methods**
> |Method|SST-2|QNLI|
> |---------------|-------|------|
> |TS-DTP (2025)|91.9|86.1|
> |SOA (2025)|91.3|89.2|
> |APS (2024)|92.0|89.4|
> |AD-TP6|92.0|89.5|
> |AD-TP12|94.7|90.2|
>
> ***W3: Regarding training details***
>
> We thank the reviewer for raising this important point and fully agree that detailed training information is crucial for understanding and reproducibility.
>
> To support reproducibility, we have provided full implementation details in Appendix A.3 of the main paper. All experiments use PyTorch and Huggingface Transformers on a single **NVIDIA RTX 3060 GPU**. The teacher is a fine-tuned BERT-base (12 layers, 768 hidden size, 12 heads), trained for **5 epochs** with a **2e-5 learning rate** and **batch size 32**. The student model uses **6 or 12 layers**, with other settings unchanged. **Learning rates** and **loss weights** (α, β, γ) are tuned over predefined grids. The candidate pruning ratios are  **$L_p = \\{0.1, 0.2, ..., 1.0\\}$**.  The detailed task-specific configurations are provided in Table R2.
>
> **Table R2: Parameter Settings for Different Datasets**
> |Dataset|Learning Rate|Epochs|Batch Size|α|β|γ|
> |---------|----------------|--------|-------------|-----|-----|-----|
> |CoLA|2e-5|20| 32|0.9|50|10|
> |MNLI|3e-5|30|16|0.8|50|500|
> |SST-2|2e-5|30|32|0.8|10|100|
> |QNLI|2e-5|30|32|0.9|100|100|
> |MRPC|3e-5|30|16|0.9|100|100|
> |QQP|4e-5|30|16|1.0|100|500|
> |RTE|2e-5|30|16|0.9|10|500|
> |STS-B|5e-5|10|16|0.8|10|100|
>
> Our method is built on **BERT-base**. The AD-TP modules (IG, ARP, TSP) are jointly trained end-to-end on each **downstream task** (e.g., SST-2, SQuAD) without extra pretraining or external data.
>
> Per your suggestion, we also report **training time statistics** for each GLUE task (see Table R3), and will further highlight them in the revised version.
>
> **Table R3: Training Time Statistics for GLUE Tasks**
> |Dataset|AD-TP Time (h)|
> |---------|----------------|
> |CoLA|1.17|
> |MNLI|40.50|
> |SST-2|4.33|
> |QNLI|8.67|
> |MRPC|0.33|
> |QQP|28.83|
> |RTE |0.20|
> |STS-B|0.25|
>
> ***W4: Regarding the evaluation of TSP***
>
> Thank you for raising this important question. TSP is specifically designed to imitate IG's saliency behavior. It is trained under the guidance of IG scores from both the teacher and student models, allowing it to accurately reflect IG's token-level importance assessments.
>
> In addition, to evaluate its effectiveness, we randomly sampled 1,000 examples from the validation sets of SST-2 and MRPC, and conducted a **quantitative analysis** using the following two metrics:
>
> - **Spearman ρ:** Measures the consistency in token importance rankings between TSP and IG.
> - **MSE:** Measures the numerical similarity between the importance scores predicted by TSP and those from IG.
>
> **Table R4: Consistency Between TSP and IG Scores**
> |Dataset|Spearman ρ|MSE|
> |---------|-------------|--------|
> |SST-2|0.812|0.0119|
> |MRPC|0.778|0.0143|
>
> These results show that TSP achieves **high ranking consistency** (ρ > 0.75) and **low numerical error** with respect to IG, demonstrating that it can effectively learn and approximate the saliency distribution derived from IG.

---

> > ### Comment · Reviewer_joVc · 2025-08-05
> >
> > Thank you for your response. The rebuttal clarifies quite a bit of the methodological concerns I had regarding the setup + efficacy of individual components. I still stand by my original assessment that the original manuscript is difficult to read, replicate and draw concrete implementations from, but the rebuttal is a good start in the right direction. I have adjusted my score accordingly and increased it.

---

> > > ### Author Response · Authors · 2025-08-05
> > >
> > > Thank you very much for your thorough review and feedback. We are glad that our response was able to address some of your concerns regarding the methodology and the effectiveness of individual components. In the revised version, we will focus on optimizing the structure and clarity, and provide more explicit steps and details so that readers can more easily understand and reproduce our work. We sincerely appreciate your increased score and the opportunity to further improve the paper.

---

### Note · Authors · 2025-08-12

We sincerely thank all reviewers and the Area Chair for their thorough evaluation, constructive discussions, and recognition of our contributions. Across the four reviews, we are encouraged by the positive assessments regarding the **novelty, soundness, and practical value** of this work.

Several reviewers commended our **principled use of Integrated Gradients** for token importance estimation, acknowledged the **instance-level adaptive pruning ratio**, the intuitive division of “how many” and “which” tokens via the **ARP and TSP** modules, and the empirical results demonstrating improvements in both efficiency and accuracy. We are particularly encouraged that multiple reviewers characterized our method as **“technically sound,” “methodologically robust,” “novel and well-justified,” and “a significant and well-justified step forward.”**

We are pleased to see that, through the rebuttal and additional experiments, we **addressed key issues** including more in-depth ablation analyses, cross-architecture generalization, and practical efficiency evaluations. Notably, all reviewers, after acknowledging these clarifications, **gave more favorable overall evaluations**, with several explicitly stating that they had **raised their ratings** or maintained **clear acceptance recommendations**. This indicates that the core ideas and results of AD-TP have been **fully understood** and **highly recognized**.

Once again, we sincerely thank the reviewers and the Area Chair for the time, effort, and thoughtful reassessment they have invested in the review process. We hope our detailed responses and the constructive dialogue have further reinforced the clarity, significance, and impact of this work.

---

### Decision · Program_Chairs · 2025-09-17

**Decision:**

Accept (poster)

**Comment:**

The paper provides a method for adaptively pruning tokens to reduce the computational burden of transformers. The reviews provided positive feedback on the high-level idea and its implementation. They mentioned it to be novel, e.g., tGep “The paper's core ideas (use of Integrated Gradients (IG) for supervising token importance) are novel and well-justified”, QJNG “The core innovation of using Integrated Gradients (IG) instead of attention scores as the basis for pruning is a significant and well-justified step forward.”, and well-motivated (tGep “The architecture of the Adaptive Token Retainer, split into a ratio predictor (ARP) and a saliency predictor (TSP), is intuitive...”, vH7T “the idea of estimating sequence-level difficulty and assigning an optimal pruning ratio for each sequence is technically sound.”). In terms of the empirical bottom-line results, the reviews were overall positive, e.g., tGep and QJNG mentioned they were impressed by the improvement over the baselines. One review (joVc) did mention the benchmarks were somewhat dated, but this issue was resolved during the rebuttal when the authors provided additional experiments on more recent benchmarks with additional baseline methods.

One weakness that was not fully resolved is that of the writing quality. Both vH7T and joVc had several comments about unclear details in the paper making it difficult to both understand and reproduce the results. The authors provided clarifications in the rebuttal that seem to mostly mitigate the issues, but these should be integrated into the paper for its final version.

To conclude, the main weakness of clarity seems to me to be possible to solve without another round of reviews. Considering the mentioned positive aspects of the paper, I believe it will be a welcome addition to NeurIPS.